# Characterization of Purple Carrot Germplasm for Antioxidant Capacity and Root Concentration of Anthocyanins, Phenolics, and Carotenoids

**DOI:** 10.3390/plants12091796

**Published:** 2023-04-27

**Authors:** María Belén Pérez, Sofía Carvajal, Vanesa Beretta, Florencia Bannoud, María Florencia Fangio, Federico Berli, Ariel Fontana, María Victoria Salomón, Roxana Gonzalez, Lucia Valerga, Jorgelina C. Altamirano, Mehtap Yildiz, Massimo Iorizzo, Philipp W. Simon, Pablo F. Cavagnaro

**Affiliations:** 1Consejo Nacional de Investigaciones Científicas y Técnicas (CONICET), Av. Rivadavia 1917, Ciudad Autónoma de Buenos Aires C1425, Argentina; 2Instituto Nacional de Tecnología Agropecuaria (INTA), Estación Experimental Agropecuaria La Consulta, Ex Ruta 40 km 96, La Consulta M5567, Argentina; 3Departamento de Química, Facultad de Ciencias Exactas y Naturales, Instituto de Investigaciones Físicas de Mar del Plata, Universidad Nacional de Mar del Plata (CONICET-UNMDP), Mar del Plata B7602, Argentina; 4Instituto de Biología Agrícola de Mendoza (IBAM), Consejo Nacional de Investigaciones Científicas y Técnicas (CONICET), Facultad de Ciencias Agrarias, Universidad Nacional de Cuyo, Almirante Brown 500, Chacras de Coria M5528, Argentina; 5Instituto Argentino de Nivología, Glaciología y Ciencias Ambientales (IANIGLA, CONICET-UNCuyo-Gobierno de Mendoza), Av. Ruiz Leal s/n, Parque General San Martín, Mendoza M5500, Argentina; 6Facultad de Ciencias Exactas y Naturales, Universidad Nacional de Cuyo, Padre Jorge Contreras 1300, Mendoza M5500, Argentina; 7Department of Agricultural Biotechnology, Faculty of Agriculture, Van Yüzüncü Yıl University, 65080 Van, Turkey; 8Department of Horticultural Science, North Carolina State University, 2721 Founders Drive, Raleigh, NC 27695, USA; 9Plants for Human Health Institute, North Carolina State University, 600 Laureate Way, Kannapolis, NC 28081, USA; 10Department of Horticulture, University of Wisconsin-Madison, United States Department of Agriculture-Agricultural Research Service (USDA-ARS), Vegetable Crops Research Unit, 1575 Linden Drive, Madison, WI 53706, USA; 11Instituto de Horticultura, Facultad de Ciencias Agrarias, Universidad Nacional de Cuyo, Almirante Brown 500, Chacras de Coria M5528, Argentina

**Keywords:** *Daucus carota*, anthocyanins, phenolic compounds, carotenoids, antioxidant capacity

## Abstract

The present study characterized a genetically and phenotypically diverse collection of 27 purple and two non-purple (one orange and one yellow) carrot accessions for concentration of root anthocyanins, phenolics, and carotenoids, and antioxidant capacity estimated by four different methods (ORAC, DPPH, ABTS, FRAP), in a partially replicated experimental design comprising data from two growing seasons (2018 and 2019). Broad and significant (*p* < 0.0001) variation was found among the accessions for all the traits. Acylated anthocyanins (AA) predominated over non-acylated anthocyanins (NAA) in all the accessions and years analyzed, with AA accounting for 55.5–100% of the total anthocyanin content (TAC). Anthocyanins acylated with ferulic acid and coumaric acid were the most abundant carrot anthocyanins. In general, black or solid purple carrots had the greatest TAC and total phenolic content (TPC), and the strongest antioxidant capacities, measured by all methods. Antioxidant capacity, estimated by all methods, was significantly, positively, and moderately-to-strongly correlated with the content of all individual anthocyanins pigments, TAC, and TPC, in both years (r = 0.59–0.90, *p* < 0.0001), but not with the carotenoid pigments lutein and β-carotene; suggesting that anthocyanins and other phenolics, but not carotenoids, are major contributors of the antioxidant capacity in purple carrots. We identified accessions with high concentration of chemically stable AA, with potential value for the production of food dyes, and accessions with relatively high content of bioavailable NAA that can be selected for increased nutraceutical value (e.g., for fresh consumption).

## 1. Introduction

Carrots (*Daucus carota* L.) are widely grown and consumed worldwide; they are among the top 10 vegetables, based on global production records [1]. The species harbors a broad range of phenotypic variation, which has been exploited for decades by breeders, resulting in cultivars varying for many agricultural, physiological, and consumer quality traits. Among the latter, materials with different root colors, owing to the accumulation of different carotenoid and anthocyanin pigments, are increasingly available in vegetable markets. They include purple, yellow, red, white, and the typical orange-rooted carrots, as well as carrots with different combinations of these pigments. Black or solid purple carrots are rich in anthocyanins, which are water-soluble flavonoids, a sub-class of phenolic compounds. Among the root phenotypes rich in carotenoids (carotenoids are lipid-soluble 40-carbon isoprenoids), orange carrots mainly accumulate α- and β-carotene, both provitamin A carotenoids (PACs); yellow carrots have high levels of xanthophylls, with predominance of lutein; and red carrots are rich in lycopene. White-rooted carrots present nearly undetectable levels of the pigments found in yellow, orange, and purple carrots [2]. 

In general, the consumption of both pigment types, carotenoids and anthocyanins, has been associated with a reduced risk of chronic diseases, including some types of cancers, cognitive decline, neurodegenerative and cardiovascular diseases, and age-related macular degeneration, among others [3,4]. Although the exact mechanisms by which anthocyanins and carotenoids exert these protective effects have not been clearly identified, their health benefits have been consistently associated with their strong antioxidant capacity, presumably by means of reducing reactive oxygen species and, thereby, attenuating oxidative stress-induced damage in cell components [3,4]. In addition, anti-inflammatory and antiproliferative effects have been reported for anthocyanin and carotenoid pigments. Despite the general health benefits attributed to both of these pigment groups, previous studies from diverse plant species, used as pigment sources, have demonstrated that individual carotenoids and anthocyanins have specific and distinct physicochemical properties, bioavailability, bioactivities, and health-enhancing attributes [3,4,5,6]. This suggests that carrots varying in pigment content and composition will also vary in their value as functional foods. In addition to pigment variation, carrot cultivars vary in their composition and content of other phytochemicals, including non-anthocyanin phenolic compounds, polyacetylenes and terpenes, vitamins, etc. [7], which may contribute to different extents to carrots’ nutraceutical properties.

The purple carrot germplasm harbors ample phenotypic variation with regards to pigment content and composition, both for anthocyanins and carotenoids, exhibiting also different patterns of pigment distribution across the root tissues [8]. Anthocyanins can accumulate in high quantities across all the root tissues, resulting in solid dark purple or black carrots; or they can accumulate in varying concentrations in specific tissues like the periderm, the outer phloem (also called ‘cortex’), the inner phloem, and the xylem, or in different combinations of the latter, exhibiting different patterns of root tissue-specific anthocyanin pigmentation. Such purple pigmentation patterns are usually combined with different carotenoid pigments, most often orange (with α- and β-carotene) and yellow (predominantly lutein) carotenoids, although purple roots with white (having traces or no carotenoids at all) and more rarely red (due to lycopene) color in the background can also be found. Such a diverse array of pigment combinations makes purple carrots an interesting food source for delivering water- and lipid-soluble phytonutrients to the consumer, and justifies the characterization of these germplasm with regards to nutraceutical value, such as their antioxidant capacity. 

Anthocyanin consumption, in particular, has been associated with various health benefits, including a lower incidence of cardiovascular disease, diabetes, arthritis, and some types of cancers, as well as promoting brain function; and such effects have been attributed to the antioxidant and anti-inflammatory properties of these pigments [4,9,10]. The few studies published to date describing the purple carrot germplasm indicate a broad variation of root anthocyanin concentration, reporting altogether an overall range for total anthocyanin content of ~5–1910 mg kg^−1^ fw [11,12,13]. These results from previous studies may suggest that a comparable broad variation for nutraceutical value, e.g., for antioxidant capacity, exists in the purple carrot germplasm associated with root anthocyanin levels; however, such a hypothesis has not been evaluated to date.

Concerning anthocyanin composition, purple carrots accumulate almost exclusively cyanidin glycosides, with five major compounds reported in most of the accessions analyzed; three of them acylated and two non-acylated [11,14,15,16]. In general, acylated anthocyanins (AA) predominate over non-acylated anthocyanins (NAA), with the former accounting for 55–99% of the total anthocyanin content [11,12,13]. According to Kammerer et al. [11], cyanidin glycosides acylated with ferulic (Cy3XFGG), sinapic (Cy3XSGG), and coumaric acid (Cy3XCGG) are, in that order, the most abundant pigments found in purple carrot roots. 

The composition of anthocyanins in the purple carrot is highly relevant, as these pigments are used as colorants for the food industry, and it has been found that AA are chemically more stable (i.e., they are less susceptible to degradation under a broader range of temperature, light, and pH conditions used for conservation of food products) than their non-acylated counterparts, suggesting that the former are more suitable as food dyes [9,15,17]. Furthermore, a recent study demonstrated that among carrot AA, those acylated with ferulic (Cy3XFGG) and coumaric acid were the most chemically stable carrot pigments [18]. Conversely, NAA were found to be substantially more bioavailable than AA, in studies using anthocyanins from carrot [14,19] and red cabbage [20]. Considering that bioavailability (i.e., the fraction of a nutrient in food that is absorbed and utilized) is a major component of a compound’s nutraceutical value, from a health-promoting perspective, a higher ratio of NAA:AA would be desirable in purple carrots for fresh consumption. On the other hand, a high AA:NAA ratio would be ideal for the production of chemically stable food dyes. Such divergent market end-purposes associated with anthocyanin composition warrant the characterization of the purple carrot germplasm for anthocyanin content and composition. 

Carrots of all root colors accumulate non-anthocyanin phenolics, predominantly those with a single aromatic ring known as phenolic acids [2]. The main phenolic acid found in carrot roots is chlorogenic acid [21]. Carrot polyphenols have been reported to have potent antioxidant and free-radical scavenging properties that may protect against oxidative damage to important biomolecules, as well as anti-inflammatory effects, both of which may contribute to reducing the risk of cardiovascular and neurodegenerative disorders, and some cancers [22]. 

Previous studies have shown that the level and composition of anthocyanins and phenolics in many fruit and vegetable crops can be strongly influenced by environmental factors, agricultural practices, and a number of abiotic stresses [23]. For example, changes in environmental conditions, such as increased temperature during the crop cycle, can influence total anthocyanin concentration and pigment composition, particularly with regards to the proportion of acylated anthocyanins in some species [24,25]. This suggests that the evaluation of the purple carrot germplasm for these environmentally influenced traits should ideally be carried out across different environments. However, to date, all the studies that have characterized anthocyanin composition in purple carrots were carried out in a single growing location and year (i.e., a single genetic environment) [11,12,13]. Additionally, these previous studies evaluated relatively few accessions (2–15) for anthocyanin composition, but other major carrot phytochemicals (e.g., phenolics and carotenoids) or their antioxidant capacities were not assessed.

Thus, in the present study, we characterized root anthocyanin concentration and composition, total phenolics, β-carotene and lutein, as well as antioxidant capacity (by DPPH, ABTS, FRAP, and ORAC) in a collection of 27 purple and two non-purple (one orange and one yellow) carrot accessions, using a partially-replicated two-year experimental design of field-grown carrots, with the purpose of the following aims.

(i) Characterize the purple carrot germplasm for these variables under different environments or growing seasons; 

(ii) Identify accessions with high concentration of chemically stable AA (for the production of food dyes) or bioavailable NAA (for fresh consumption), as well as materials with high antioxidant capacity, suitable for both purposes; 

(iii) Estimate the environmental influence on these traits; 

(iv) Investigate relationships between the level of phytochemicals and antioxidant capacity.

## 2. Results

Root phenotypes for the 29 carrot accessions evaluated in this study are presented in Figure 1. In addition, Table 1 presents data concerning the accessions names and IDs at germplasm banks, their genetic structure, petiole and root phenotypes, seed source, and geographic origins.

### 2.1. Variation for Anthocyanin Content and Composition among Carrot Accessions 

Data for total anthocyanin content, as estimated by spectrophotometry analysis (TAC_SPEC_), in the carrot accessions with purple roots (the solid yellow and orange-rooted accessions were not assessed for TAC_SPEC_) for 2018 and 2019 are presented in Figure 2. In general, considering all the accessions combined, the mean TAC_SPEC_ values were significantly greater (*p* = 0.0016) in 2019 than in 2018 (overall means were 293.8 vs. 238.4 mg kg^−1^ fw). The range of mean TAC_SPEC_ values found among the accessions was 1.5–1087.0 and 7.7–1633.1 mg kg^−1^ fw for 2018 and 2019, respectively. Accession 1 had the greatest TAC_SPEC_ level in both years, whereas accessions 1–6 and 1–7 [corresponding to the cultivars or breeding lines P9547, Purple 68, Pusa asita, Night Bird, INTA43, Black nebula, and Black carrot (Table 1)] had significantly higher TAC_SPEC_ levels than the rest of the accessions for years 2018 and 2019, respectively, although their relative rank order varied between years.

The root total anthocyanin concentration was also estimated by means of HPLC-UV-Vis analysis (TAC_HPLC_) in the same 27 anthocyanin-containing accessions. Results for both years of analysis are presented in Figure 3. Overall, no significant differences were found between years, whereas broad and significant variation was observed for mean TAC_HPLC_ values among the accessions, namely 1.5–3014 and 0.8–3330.1 mg kg^−1^ fw for 2018 and 2019, respectively. In general, the accessions rank order for TAC_HPLC_ coincided with that of TAC_SPEC_, with accession 1 (P9547) presenting the greatest anthocyanin concentration, and accessions 1–8 having significantly greater TAC_HPLC_ levels than the rest of the accessions in both years. The total anthocyanin content, as estimated by both spectrophotometry and HPLC-UV-Vis analyses, was associated with the root color phenotype, that being the accessions with black or solid purple roots—i.e., roots intensively pigmented across all tissues (e.g., accessions 1–5)— the ones with greatest anthocyanin concentration, whereas accessions exhibiting purple pigmentation only in the periderm (e.g., accessions 18–26) had the lowest pigment levels (Figure 1, Figure 2 and Figure 3, Table 1). Conversely, purple pigmentation in the leaf petioles was not associated with root anthocyanin concentration, with it being the former leaf phenotype present in accessions with very high (accessions 1–7) and very low root anthocyanin content (e.g., accessions 18–27).

Anthocyanin composition by HPLC-UV-Vis analysis was examined in 27 purple-rooted accessions. Five major anthocyanins pigments, three acylated (Cy3XSGG, Cy3XFGG, and Cy3XCGG), and two non-acylated (Cy3XG and Cy3XGG), were identified in most of the accessions (Figure 4, Table 2). The proportion of acylated anthocyanins (AA), relative to the total anthocyanin content, ranged from 54.3 to 98.8%, in 2018, and 56.8 to 100% in 2019, with most of the accessions having more than 80% of AA (Figure 4 and Appendix A). Accession 5 (INTA43) had the lowest percentage of AA in both years (55.6% on average). Conversely, accessions 8 (B7262), 11 (Purplesnax), and 13 (Purple elite) had the greatest AA percentage in 2018 (all above 97%), whereas three carrot landraces, corresponding to accessions 17 (PI 223361), 25 (PI 254552), and 26 (PI 226636), exhibited anthocyanin profiles with only acylated pigments (i.e., 100% AA). The cyanidin glycoside acylated with feruloyl, Cy3XFGG, was the most abundant pigment in 21 of the accessions, with its mean content for both years representing 50.3–100% of the total anthocyanin content in these accessions, and 10.9–100% in the entire collection of carrot materials. Another AA, acylated with sinapoyl, Cy3XSGG, predominated in the other 6 accessions, and accounted for 32.2–100% of total anthocyanins in this subset of accessions, and 0–100% in the entire collection. Non-acylated pigments all combined represented a smaller fraction of the total anthocyanin content, varying among the accessions from 0% (in accessions 17, 25, and 26) to 44.5% (in accession 4; INTA43). Among the NAA, Cy3XG predominated over Cy3XGG in 19 of the 27 accessions. Overall, the pigments with the least relative abundance in all the accessions were Cy3XCGG (0.0–9.7%) and Cy3XGG (0.0–12.6%). 

### 2.2. Variation for Total Phenolic Content 

The root total phenolic content (TPC) was estimated for 29 carrot accessions. Figure 5 depicts the results for 2018 and 2019. Overall, no statistical differences were found between years, whereas significant and substantial variation was revealed among the accessions. Mean TPC values varied ~20 and 28 folds across the accessions, with ranges of 172–4669 and 215–4310 mg GAE kg^−1^ fw for 2018 and 2019, respectively. In general, black or solid purple carrots and accessions with visually intense purple coloration (accessions 1–7; Figure 1) had the greatest TPC levels in both years, with accession 1 (P9547) being significantly richer in total root phenolics than the rest.

### 2.3. Carotenoids Content

The root contents of two major carotenoids, β-carotene and lutein, for the carrot accessions grown in 2018 are presented in Figure 6. Significant and substantial differences were found among the accessions for both pigments. β-carotene levels were on average for the entire collection ~7 folds greater than lutein levels, with overall means and ranges of 28.6 and 0.5–102.1, and 4.1 and 1.3–9.7 µg g^−1^ fw, for β-carotene and lutein, respectively. The solid orange (accession 29) and a carrot with purple periderm and orange phloem and xylem (accession 21) had the greatest mean β-carotene contents, whereas a fully yellow-rooted carrot (accession 28) had the greatest lutein content (Figure 1 and Figure 6). In the solid-purple carrots (accessions 1–5), the level of these carotenoids was rather low, exhibiting ranges of 2.0–7.2 and 0.6–4.7 µg g^−1^ fw, for β-carotene and lutein, respectively.

### 2.4. Antioxidant Capacity 

#### 2.4.1. ORAC

Antioxidant capacity, as estimated by the ORAC assay, in the carrot accessions grown in 2018 revealed significant differences, with a range of variation of ~6 folds, considering the accessions with weakest (accession 18) and strongest capacities (accession 6) (Figure 7). In general, accessions with solid purple or intense purple coloration in their roots (e.g., accessions 1–7, Figure 1) had the highest mean antioxidant values. Conversely, the lowest antioxidant capacities were found in carrots with little or no anthocyanin pigmentation, in which purple pigmentation, when present, was generally restricted to the root periderm tissue (accessions 18–29). 

#### 2.4.2. DPPH

Antioxidant capacity by DPPH in the carrot accessions grown in 2018 revealed significant and broad variation, exhibiting a ~22-fold difference between the two most contrasting accessions (Figure 8). Coincidently with results by ORAC, greater DPPH mean values were found in carrots with intense root purple pigmentation, with accessions 1–7 exhibiting significantly greater antioxidant capacities than the rest, particularly accessions 1 and 2 (P9547 and Purple 68) which had the strongest activities of all. In contrast, and coincidently with results by ORAC, carrots with little or no anthocyanin pigmentation presented the lowest antioxidant capacities (accessions 16–29). 

#### 2.4.3. ABTS

Figure 9 presents data for antioxidant capacity by ABTS in the carrot accessions grown in 2018 and 2019. Overall, ABTS mean values were significantly higher (*p* < 0.0001) in 2018 than in 2019 (81.1 vs. 49.3 mmol Trolox kg^−1^ fw). Significant variation among the accessions was found for both years, with a range of values spanning a ~11-fold difference between the two most contrasting accessions. Accession 1 (P9547) had a significantly greater antioxidant capacity than the rest of the accessions in both years. In general, and coincidently with results by ORAC and DPPH, carrots with intense purple pigmentation in their roots (accessions 1–7) tended to have stronger antioxidant capacities than carrots with little or no purple pigmentation (accessions 18–26). 

#### 2.4.4. FRAP

Figure 10 presents data for antioxidant capacity by FRAP in the carrot accessions grown in 2018 and 2019. Overall, FRAP mean values were significantly higher (*p* < 0.0001) in 2018 than in 2019 (713.9 vs. 184.4 mmol Trolox kg^−1^ fw). In addition, a significant and broad variation was found among the accessions for both years, with this analytical method revealing the greatest range of values for antioxidant capacity among the taxa, varying ~238 and ~198 folds between the most contrasting materials in 2018 and 2019, respectively. Carrots with intense anthocyanin pigmentation in their roots (accessions 1–7) had significantly greater antioxidant capacity than the rest of the accessions.

### 2.5. Relationships between Bioactive Compound Content and Antioxidant Capacity 

Pairwise correlation analysis among all the variables revealed, in general, highly significant (*p* < 0.001), positive, and strong correlations between the content of all the individual, combined, and total anthocyanins, regardless of the method of analysis, as well as total phenolic content (TPC), with antioxidant capacity measured by all four methods, in both years, with ranges of correlation values of 0.64–0.90 and 0.59–0.87 for 2018 and 2019, respectively (Table 3). In contrast, β-carotene levels were either not significantly correlated (with DPPH) or they were significantly, weakly, and negatively correlated with antioxidant capacity (for FRAP, ABTS, and ORAC) with r values ranging from −0.36 to −0.49. Lutein content was not significantly associated with any other variable. TPC was strongly correlated with all measurements of individual or combined anthocyanins in 2018 (r = 0.81–0.94) and 2019 (r = 0.79–0.88), suggesting that these flavonoid pigments represent a large proportion of the total phenolics content of purple carrots. 

The different methods for estimating antioxidant capacity were also significantly and positively correlated, with r values among all four methods in the range of 0.66–0.87 for 2018; whereas in 2019 a moderate correlation was found between ABTS and FRAP (r = 0.64). Furthermore, the two methods used for estimating the total anthocyanins, namely HPLC-UV-Vis and spectrophotometry, were strongly correlated (r = 0.94–0.96).

The structure of covariance among the variables was further examined in three sub-classes of accessions varying in root anthocyanin pigmentation, as follows: (i) accessions with high pigmentation [anthocyanin concentration > 1000 mg kg^−1^ fw, typically with solid black or intense purple coloration in the periderm and the entire phloem; accessions 1–7]; (ii) accessions with intermediate pigmentation [anthocyanin concentration in the range of 80–1000 mg kg^−1^ fw, with roots exhibiting moderate purple coloration distributed mostly in the periderm and outer phloem, but sometimes also in the xylem; accessions 8–17]; and (iii) accessions with low pigmentation [anthocyanin concentration < 80 mg kg^−1^ fw, with roots exhibiting purple pigmentation mainly in the periderm and the phloem outermost cell layers, or no pigmentation at all; accessions 18–29]. Results for each of these classes, per year, are presented in Appendix A. In general, significant and positive correlations were found between TPS and anthocyanin levels with antioxidant capacity in the three groups of accessions and both years, although correlation values were generally weaker than observed for the entire germplasm collection. Additionally, a lack of significant correlation with antioxidant capacity was occasionally found for some anthocyanin pigments. Overall, considering only the significant associations found between TPC and anthocyanins with antioxidant capacity, ranges of ‘r’ values of 0.35–0.68, 0.26–0.83, and 0.17–0.68 were found for the ‘high’, ‘intermediate’, and ‘low-pigment’ accessions, respectively, in 2018; and 0.30–0.57, 0.23–0.81, and 0.23–0.37, in 2019. Interestingly, in the three subgroups of accessions and both years, neither lutein nor β-carotene levels were significantly correlated with antioxidant capacity. Altogether, these results suggest that the covariances among the variables were slightly –to moderately influenced in different sub-classes of accessions and years, but the general positive associations found between anthocyanin and phenolic levels with antioxidant capacity were consistent across analyses. 

Additional comparative analysis among these three sub-classes of accessions using ‘Cohen’s d’ statistic, which is not influenced by sample size, revealed large (d > 0.80) to very large (d > 1.30) effect size estimates among the classes for all the variables in both years, except for lutein (d = 0.06–0.27) and β-carotene (d = 0.26–2.20) in 2018 (Appendix A). These results indicate a clear differentiation among the three subgroups of accessions, as well as a strong relationship between anthocyanin content and antioxidant capacity. 

### 2.6. Principal Component Analysis

The principal component analysis (PCA) with 16 variables, including the root content of 12 bioactive compounds or compound groups, and antioxidant capacity estimated by 4 analytical methods, was conducted with combined data from years 2018 and 2019 for 29 carrot accessions (Figure 11). Two principal components (PC) were generated, together explaining 80.5% of the total variation, with PC1 accounting for 71.6% of the variation. The variables that contributed most to PC1 were, in decreasing order, TAC_HPLC_, TAC_SPEC_, TAA, TPC, FRAP, ABTS, Cy3XG, TNAA, DPPH, ORAC, Cy3XGG, Cy3XFGG, and Cy3XCGG; whereas lutein and β-carotene content contributed the most to explaining PC2. Antioxidant capacity by all four methods used was strongly and positively correlated with TPC, TAC_HPLC_, TAC_SPEC_, TAA, TNAA, and the individual anthocyanins Cy3XG, Cy3XFGG, and Cy3XCGG, and to a lesser degree with Cy3XGG and Cy3XSGG. Accessions 1–7 (located in the right half of the bi-plot) were the most representative of these variables, with accession 1 (P9547) showing the strongest association. Conversely, the β-carotene and lutein contents were not associated with antioxidant capacity. 

## 3. Discussion

In the present work, a total of 27 purple-rooted accessions and two non-purple carrots (one orange and one yellow) were characterized for root concentration of anthocyanins, phenolics, and carotenoids, and antioxidant capacity estimated by four different methods, in a partially replicated experimental design comprising data from two growing seasons. This genetically and phenotypically diverse carrot collection comprised open-pollinated and hybrid commercial cultivars, landraces, and plant introductions from germplasm banks, and lines from the carrot breeding programs at the USDA (USA) and INTA (Argentina) (Table 1). Overall, 16 variables (12 bioactive compounds, or groups of compounds, and four estimates of antioxidant capacity) were examined on 29 accessions over two years. Thus, although a few previous studies have also evaluated purple carrots, mainly for anthocyanin and phenolic content [11,12,13,26], the present work represents the most comprehensive study published to date with regards to number of accessions, genetic environments (years), and variables analyzed. 

One of the main interests in carrot anthocyanins is their use as food colorants. For this purpose, the high total anthocyanin concentration, particularly of the acylated forms, is highly relevant, as it has been shown in previous studies that AA are chemically more stable than NAA (reviewed by Cavagnaro et al. [8], Prior and Wu [9], and Iorizzo et al. [17]). In the present study, ranges of total anthocyanin concentration of 1.5–3014 and 0.8–3330 mg kg^−1^ fw, as estimated by HPLC-UV-Vis analysis, were found for 2018 and 2019, respectively. These range values are larger than reported in previous studies with purple carrots. For example, Montilla et al. [12] analyzed four commercial cultivars and reported a range of 15–177 mg kg^−1^ fw. Kammerer et al. [11] evaluated 15 accessions, reporting a range of 45–17,400 mg kg^−1^ dry weight (dw), which corresponds to ~5–1910 mg kg^−1^ fw (considering a dry matter content of 11%), whereas Algarra et al. [13] analyzed two cultivars from Spain reporting 934–1264 mg kg^−1^ fw. More recently, five purple carrot breeding lines (within a collection of 11 lines of different root color) were analyzed, reporting total anthocyanin concentrations of 158–1477 mg kg^−1^ fw [26]. The broader range of variation for total pigment content found in the present work, as compared to previous studies, is likely due to the larger and probably more genetically diverse germplasm collection evaluated herein. In addition, the different environments where the accessions were grown may have contributed to these differences. Of the materials analyzed, accessions 1–7 exhibited a high anthocyanin concentration (with mean TAC_HPLC_ values, for both years, in the range of 1402–3172 mg kg^−1^ fw), of which accessions 1 (P9547; 2838.9 mg kg^−1^ fw), 3 (Pusa asita; 1541.1 mg kg^−1^ fw), 2 (Purple 68; 1491.5 mg kg^−1^ fw), 4 (Night bird; 1373.2 mg kg^−1^ fw), and 7 (Black carrot; 1322 mg kg^−1^ fw) had the greatest AA concentration, in absolute values, suggesting that these materials will be highly suitable for the production of food dyes. Furthermore, a recent study demonstrated that among carrot AA, those acylated with ferulic (Cy3XFGG) and coumaric acid (Cy3XCGG) were the most chemically stable anthocyanins [18]. Accessions 1, 2, 4 and 7, which were all rich in AA, had the highest Cy3XFGG content, with mean values for both years ranging from 1096 to 1866 mg kg^−1^ fw, suggesting that these are the most suitable genotypes in our germplasm collection for extracting chemically stable food colorants. Conversely, Cy3XCGG was present in a very low concentration in all the carrot accessions analyzed, and therefore its contribution to pigment stability is rather low. 

The total anthocyanin content, as estimated by spectrophotometry and HPLC-UV-Vis analyses, varied in their ranges of absolute values (4.4–1360 vs. 1.1–3172.1 mg kg^−1^ fw; mean range values for both years), but they were highly concordant in the relative contents and rank order of the accessions for this trait (Figure 2 and Figure 3), presenting significant and strong positive correlations between both methods in both years (r = 0.94–0.96) (Table 3). According to a study by Lao et al. [27], which compared the same methods used in the present work (i.e., the pH differential method vs. HPLC-UV-Vis analysis), discrepancies in absolute values obtained by both methods are rather frequent and expected, and such quantitative differences are likely due to differences in their specificity. Nonetheless the latter, and other previous studies performing the same comparison, have reported strong positive correlations (r = 0.93–0.99) between the two methods [27,28,29]. Considering our TAC_HPLC_ data, for standardized comparative purposes with other anthocyanin-containing fruits and vegetables, the range of values found in this carrot collection is broader than generally reported for other species [30]. However, when considering the most intensively pigmented purple carrots (i.e., accessions 1–7), which exhibited a TAC_HPLC_ range of 1402–3172.1 mg kg^−1^ fw (mean for both years), it is apparent that their total anthocyanin content is lower than the TAC_HPLC_ range reported for purple currants and blueberries (3650–4760 mg kg^−1^ fw), but higher than reported for plums (190–1245 mg kg^−1^ fw), strawberries (211–417 mg kg^−1^ fw), red onion (485 mg kg^−1^ fw), and red grapes (267 mg kg^−1^ fw) [30]. 

A few of the plant materials analyzed, namely accessions 13 (Purple elite) and 14 (Purple haze), and accessions 16 (Gniff) and 17 (PI 223361), were remarkably similar with regard to root and petiole phenotypes (Figure 1 and Table 1), genetic structure (for accessions 13 and 14; Table 1), anthocyanin content, and composition (Figure 2, Figure 3 and Figure 4), and TPC, carotenoids, and antioxidant levels (Figure 5, Figure 6, Figure 7, Figure 8, Figure 9 and Figure 10). Based on this observation, we hypothesize that, in the case of accessions 13 and 14, it is possible that different seed companies commercialize the same carrot seed, obtained from a larger seed provider under different commercial names. With regards to the other pair of phenotypically and biochemically similar materials (i.e., accessions 16 and 17), one is an open-pollinated commercial cultivar and the other one is a landrace plant introduction from the GRIN-USDA germplasm bank. Although these two share similar root phenotypes (Figure 1), and statistically comparable biochemical traits (Figure 2, Figure 3, Figure 4, Figure 5, Figure 6, Figure 7, Figure 8, Figure 9 and Figure 10), accession 16 was phenotypically more uniform than accession 17 (Figure 1), suggesting that the cultivar Gniff may have been developed from PI 223361, or from a derivative population of the latter landrace. Additional studies using molecular markers are needed to conclusively determine whether these pairs of accessions correspond to synonymous genotypes or, if not, to estimate the degree of genetic relatedness between them. 

The total phenolic levels, as estimated by the Folin—Ciocalteu assay, varied significantly among the accessions, with a range of 172–4669 mg GAE kg^−1^ fw (considering both years). This TPC range is in general comparable to, yet broader than, found in previous studies with purple carrots, which have reported altogether values in the range of 179–3115 mg GAE kg^−1^ fw [12,21,31]. The broader TPC range found in the present study is likely due to the larger germplasm collection analyzed, which included some accessions with very high anthocyanin content, possibly raising the upper limit of the TPC range. The facts that TPC was strongly correlated with all measurements of individual or combined anthocyanins in both years (r = 0.79–0.94), and that carrots with low anthocyanin content also had low TPC levels, suggests that these pigments represent a large proportion of the total phenolics content, in full agreement with previous observations by Leja et al. [21].

In this work, four methods were used for estimating carrot antioxidant capacity. According to Huang et al. [32], these analytical procedures measure different components, or modes of action, of an antioxidant agent, and therefore it is ideal to use several methods to accurately predict the overall antioxidant capacity. Despite this, all the assays used revealed similar trends with regard to the relative rank order of the accessions, with dark purple carrots generally presenting the strongest antioxidant capacities (Figure 7, Figure 8, Figure 9 and Figure 10).

DPPH, ABTS, and ORAC measure the scavenging capacity against radical oxygen species, where the antioxidant transfers electrons or hydrogen atoms to the radical species [33]. Among these radical scavenging methods, ORAC is of particular biological relevance and serves as a reference for antioxidant effectiveness since it measures the radical chain breaking ability of antioxidants by monitoring the inhibition of peroxyl radical-induced oxidation [34]. Peroxyl radicals are the predominant free radicals found in lipid oxidation processes in biological systems under physiological stress conditions [35]. On the other hand, FRAP measures the electron donating capability of an antioxidant to reduce Fe^+3^ to Fe^+2^ [33]. This redox potential or reducing power of antioxidants is an important indicator of their antioxidant efficacy. Our results indicate that the main contributors to antioxidant capacity in purple carrots were anthocyanins and TPC. The antioxidant capacity estimates obtained by ABTS, DPPH, and ORAC were generally comparable in their range values (12.3–727.2 mmol Trolox kg^−1^ fw), whereas estimates by FRAP revealed a much broader range (11.4–2710.5 mmol Trolox kg^−1^ fw). These results suggest high antioxidant efficacy for carrot anthocyanins and other phenolics, presumably exerting their antioxidant effects by both mechanisms, radical scavenging and redox potential. Such versatility in the antioxidant mechanisms of anthocyanins and other phenolic has been thoroughly discussed in a number of studies [36,37,38].

In general, fairly comparable results were obtained between years and across different methods used for estimating both phytochemical content (e.g., TAC_HPLC_ vs. TAC_SPEC_) and antioxidant capacity (ABTS, ORAC, DPPH, FRAP), and similar relationships were established between the two datasets; i.e., bioactive compounds and antioxidant capacity for both years. In general, accessions with solid black or intensively purple-colored roots (e.g., accessions 1–7) had the greatest anthocyanin and TPC levels (considering individual and combined compounds, and regardless of method of analysis) and the strongest antioxidant capacities, often being significantly greater than the rest of the accessions (Figure 2, Figure 3, Figure 4, Figure 5, Figure 6, Figure 7, Figure 8, Figure 9 and Figure 10). In line with this observation, our results from correlation analysis and PCA (Table 3 and Figure 11) strongly suggest that anthocyanins and more broadly total phenolics are the largest contributors to the antioxidant capacity in purple carrots. However, the fact that the total anthocyanin content was strongly correlated with TPC (r = 0.86–0.94), and that carrots with low anthocyanins also had low TPC (Figure 2, Figure 3, Figure 4 and Figure 5), suggest that anthocyanins are accounting for most of the TPC levels, and therefore also for the strong correlation found between TPC and antioxidant capacity (r = 0.66–0.86). Furthermore, the fact that the total anthocyanins content, by TAC_HPLC_ and TAC_SPEC_, was somewhat more strongly correlated with antioxidant capacity (r = 0.72–0.90) than the content of individual anthocyanin pigments with antioxidant capacity (r = 0.64–0.78), without evident differences in the strength of such association among individual pigments, suggests that for exerting potent antioxidant effects, total anthocyanin concentration is more important than pigment composition. However, it must be noted that these determinations of antioxidant capacity were performed in vitro, without considering other factors influencing the nutritional value of a given bioactive compound, such as bioavailability. Previous studies using anthocyanins from carrots [14,19] and red cabbage [20] have demonstrated that NAA were substantially more bioavailable than AA. Therefore, from a nutritional perspective aimed at effectively improving antioxidant status by dietary means, it would be ideal to consume carrots with a high anthocyanin concentration, particularly of the non-acylated forms. Although in the present study all of the accessions revealed a predominance of AA over NAA, three accessions with a high pigment content and a relatively high proportion of NAA were identified, namely accessions 5 (INTA43), 4 (Night bird), and 1 (P9547), exhibiting anthocyanin profiles in which NAA accounted for 44.5%, 27.5%, and 23.2% (values are means of both years) of the total anthocyanin content, respectively (Figure 3, Appendix A). In absolute values, these correspond to 802.1, 733.2, and 535.4 mg kg^−1^ fw, for INTA43, P9547, and Night bird, respectively. The predominance of AA over NAA found in our carrot collection (i.e., AA represented 55.5–100% of total anthocyanins) coincides with previous reports using other purple carrot germplasms. Altogether, these previous studies reported percentages of AA, relative to the total anthocyanin content, in the range of 49.6–99.0% [11,12,13]. Interestingly, although predominance of AA over NAA has been consistently reported in nearly all of the purple carrot commercial cultivars and germplasms evaluated to date [11,12,13] including this work, individual carrots with a high proportion of NAA have been reported in mapping populations derived from experimental crosses (with NAA representing up to 90% of total anthocyanins) [15,39], suggesting that carrot cultivars with a high content of bioavailable AA can be developed. Furthermore, the genetics underlying anthocyanin acylation in carrots was recently addressed, revealing that a mutation caused by an insertion in an acyltransferase gene, *DcSCPL1*, leading to a non-functional acyltransferase, is likely responsible for the low acylation phenotype (i.e., high percentage of NAA) [39]. The discovery of this acyltransferase gene and its causal mutation conditioning different AA:NAA ratios in the carrot root, may lead to new strategies for increasing the concentration of highly bioavailable anthocyanins, increasing thereby the effectiveness of these dietary antioxidants for improving human health. Among the strategies that can be used for achieving this goal is silencing *DcSCPL1* (e.g., with CRISPR-Cas9 technology) to block anthocyanin acylation in the edited lines; or introducing the non-functional mutant *DcSCPL1* gene, by classical breeding methods, into a high-pigment carrot genetic background (e.g., P9547). Conversely, increasing the activity of *DcSCPL1* (e.g., by transgenic approaches using a strong promoter upstream of *DcSCPL1*) should result in increased levels of chemically stable AA, of value as food colorants.

Carotenoid pigments were not positively associated with antioxidant capacity in purple carrots. Lutein content was not significantly correlated with antioxidant capacity or any other variable, whereas β-carotene was often weakly and negatively correlated with antioxidant capacity (r = −0.36–−0.47) (Table 3). Similar results were found in a previous study that evaluated anthocyanin, phenolics, carotenoids, and antioxidant capacity in seven carrot lines of different root color, including two purple carrots, reporting that anthocyanins and phenolics, but not carotenoids, were significantly and strongly correlated with antioxidant capacity in the hydrophilic extracts, with correlation coefficient values for anthocyanins and phenolics in the range of 0.77–0.99 [40]. Similarly, Yoo et al. [26] examined the level of carotenes, anthocyanins, and terpenoids, and antioxidant capacity in 11 breeding lines of different root color, including five purple carrots, reporting a significant correlation only between anthocyanin content and antioxidant capacity (r = 0.975). The weak yet significant correlation found between the β-carotene content and some of the antioxidant capacity estimates may be circumstantial rather than causal (i.e., it is unlikely that β-carotene has pro-oxidant activity), as most of the dark purple carrots exhibiting the highest antioxidant capacities had a very low β-carotene, whereas carrots with little or no anthocyanin pigmentation and low antioxidant capacity were richer in β-carotene (Figure 6).

## 4. Materials and Methods

### 4.1. Plant Materials and Cultivation Conditions 

A total of 29 carrot accessions from diverse geographical origins were evaluated. Of these, 27 had purple roots, and varied in their patterns of purple pigmentation, and combinations with other carotenoid pigments, across different root tissues, and the remaining two accessions had solid orange and yellow roots, respectively (Table 1, Figure 1). These materials include landraces, open-pollinated, and hybrid commercial cultivars; inbred lines were from the national carrot breeding programs of INTA La Consulta (Mendoza, Argentina) and the USDA-ARS (Madison, WI, USA), and plant introductions from the germplasm bank of the Germplasm Resource Information Network (GRIN) of the USDA-ARS. 

The carrot accessions were cultivated in the experimental field of the Faculty of Agricultural Sciences of the National University of Cuyo, in Lujan de Cuyo, Mendoza (33°0′ S, 68°52′ W, 927 m.a.s.l.), in the 2018–2019 and 2019–2020 seasons, henceforth referred to as 2018 and 2019, respectively, using conventional agricultural practices for the crop. Briefly, carrots were sown by hand and then, when the plants had 6–8 true leaves, they were thinned to ~80–90 plants/m^2^. The crop was drip-irrigated and fertilized twice with Akaphos^®^ Violeta (Compo Expert Co., Ciudad de Buenos Aires, Argentina), containing N-P-K (13–40-13) and micronutrients, and adding a total of ~120, 360, and 120 kg per hectare of nitrogen (N), phosphorous (P2O5), and potassium (K2O), respectively. The field crop, from sowing to harvest, lasted five months. The same agricultural practices were applied in both seasons. Twenty-two accessions we grown and evaluated in both years, whereas seven accessions were cultivated in a single year, namely 2019. Data for weather conditions and edaphic parameters at the cultivation site during the carrot cultivation period (i.e., from October 15 to March 15), in both years, are presented in Appendix A. 

### 4.2. Sampling and Extraction of the Hydrophilic Fraction of Carrots 

For each plant material, three roots of similar size exhibiting the typical phenotype for the accession and showing with no sign of pathogen attack or disease were individually sampled and processed. In total, 66 roots from 22 accessions were sampled in 2018, whereas 87 roots from 29 accessions were sampled in 2019. The mid-section of the root (i.e., the middle third part of the root, as cut transversally in three sections of equal length) was used for anthocyanin and phenolics extraction, as described previously [15,18]. Briefly, the root tissues were mixed with methanol acidified with 10% formic acid, in different ratios depending on the level of pigment intensity of the carrot accession [e.g., 1:3 and 1:10 (*w*/*v*), for roots with low and high purple pigmentation, respectively]. The mixture was homogenized with a mortar and pestle, then transferred to a caramel glass bottle and incubated at 4 °C in the dark for 12 h, followed by centrifugation (11,000 rpm, 4 °C) for 15 min. Finally, the supernatant was separated and stored at −20 °C until analytical determinations were performed.

### 4.3. Spectrophotometric Analyses of Total Anthocyanins and Total Phenolics 

Total phenolics content (TPC) was estimated spectrophotometrically using the Folin reagent as described by Singleton and Rossi [41]. To this end, 50 μL of the carrot methanolic extract, or just the methanolic solvent as sample blank, were mixed with 3700 µL of distilled water and 250 μL of the Folin–Ciocalteu reagent. The mixture was stirred and, after 3 min, 1 mL of a 20 % Na_2_CO_3_ solution (*w*/*v*) was added. Subsequently, the mixture was incubated at 37 °C in the dark. After 60 min, absorbance readings at 765 nm were recorded using a Beckman DU-530 UV–vis spectrophotometer (Beckman Coulter Inc., Brea, CA, USA). A gallic acid reference compound (Sigma Aldrich, Atlanta, GA, USA) was used for calibration, and values were expressed as mg of gallic acid equivalents (GAE) per kg of fresh weight (mg GAE kg^−1^ fw). 

The pH differential method was used to determinate the total anthocyanins content (TAC) in the carrot extracts, as described earlier [42]. The absorbance of the extracts was measured at pH 1.0 and pH 4.5 at two wavelengths, 520 nm and 700 nm, to correct the haze. The absorbance difference was calculated using the following equation:
$$A={\left(A_{520}-A_{700}\right)}_{\mathrm{pH}=1.0}-{\left(A_{520}-A_{700}\right)}_{\mathrm{pH}=4.5}\;$$

The absorbance difference was used to calculate anthocyanins concentration (C, mg L^−1^) as expressed in the following equation:
$$C=\frac{\mathrm{DF}\;\times \;\mathrm{MW}\;\times 1000\times \;A}{\varepsilon \times \;d}\;$$
where DF is the dilution factor, MW is molecular weight of 449.2 g mol^−1^, *A* is the absorbance difference, ε is the molar absorptivity (26,900), and d is the cuvette length (1 cm).

For both variables, TPC and TAC, three biological replicates (i.e., three extracts, one from each of three individual roots) per accession were analyzed in both years, 2018 and 2019. For each biological replicate, two measurements (analytical replicates) were obtained.

### 4.4. Anthocyanin HLPC-UV-Vis Analysis 

High-performance liquid chromatography (HPLC) was used to determine the concentration of anthocyanin pigments in the methanolic root extracts of the 27 anthocyanin-containing carrot accessions (i.e., the solid orange and yellow-rooted accessions were not analyzed for anthocyanins), according to methods previously described [18]. HPLC analysis was carried out using a UHPLC apparatus (Shimadzu, SIL30-AC, Tokyo, Japan) equipped with a binary pump system (LC-30AD, Nexera, Shimadzu), an autosampler-injector (SIL-30AC, Nexera X2, Shimadzu), a photodiode array UV–VIS detector (SPD-M30A Nexera X2, Shimadzu), and a C18 column (3 μm, 2.1 × 150 mm, UFLC Aqueous, RESTEK). The mobile phase was distilled water acidified with 1% (*v*/*v*) formic acid as solvent A and methanol as solvent B. The gradient system was 0/5, 20/55, 21/100, 26/100, 27/5, and 40/5 (min/% solvent B). A commercial standard of cyanidin (Sigma Aldrich, Atlanta, USA) was used for quantitation purposes. Peak assignment was performed by LC-MS/MS analysis of the column eluate, which was carried out with the HPLC system described above coupled to electrospray ionization tandem mass spectrometry (HPLC-ESI-MS/MS) with an ESI-QTOF instrument model G6560A from Agilent (Santa Clara, CA, USA). Table 2 presents the five major carrot anthocyanin pigments, along with their retention times and molecular masses, identified and quantified in the present study. A cyanidin standard (Sigma Aldrich, Atlanta, GA, USA) was used for quantitation purposes, and the concentration of anthocyanin pigments was expressed as mg of cyanidin equivalents per kg fw (mg kg^−1^ fw). The total anthocyanin content was calculated as the sum of concentrations of all the individual pigments. Three biological replicates (i.e., three extracts, one from each of three individual roots) per accession were analyzed in both years. 

### 4.5. Carotenoids HPLC Analysis 

The lipophilic fraction of carrots was extracted from the same root mid-section used for the extraction of the hydrophilic fraction. A sub-sample of the root mid-section was lyophilized, and 100 mg of the lyophilized tissue was ground to a fine powder using a mortar and pestle and macerated with 1 mL of ultrapure water and 5 mL of ethanol:n-hexane (60:40, *v*/*v*). The mixture was transferred to a conical glass tube (screw cap), sonicated for 15 min, and then centrifuged for 15 min at 1344× *g* (4000 rpm). During sonication, the temperature was monitored and always kept below 30 °C, to avoid carotenoid degradation and isomerization. The solvent phase was collected and transferred to another glass tube and then evaporated until dry by vacuum centrifugation. Two additional extractions were carried out by adding 3 mL aliquots of n-hexane, each one to the conical tube containing the pellet of the previously extracted sample and repeating the same process described above. All solvent phases from the same sample were collected and dried in the same glass tube by vacuum. Once the aliquots of the extract were dried, the resulting precipitate was re-suspended with a mixture of methanol:MTBE (1:1, *v*/*v*) and injected in the HPLC-DAD [43]. 

Target compounds were determined using an HPLC-DAD system (Dionex Softron GmbH, Thermo Fisher Scientific Inc., Germering, Germany). The instrument was a Dionex Ultimate 3000 comprising a vacuum degasser unit, an autosampler, a quaternary pump, and a chromatographic oven. The detector used was a Dionex DAD-3000, consisting of an analytical flow cell set to scan from 200 nm to 500 nm, operated with a data collection rate of 5 Hz, bandwidth of 1 nm, and a response time of 1.000 s. The wavelengths selected for the quantification of analytes were: 445 nm and 450 nm, for lutein and β-carotene, respectively. The control of all the system acquisition parameters and data processing were performed with the Chromeleon 7.1 software. Chromatographic separations were carried out in an Accucore C_30_ column (3.0 mm × 150 mm, 2.6 µm) from Thermo Scientific (Bellefonte, PA, USA) including an Accucore C_30_ guard column (10 mm × 2.1 mm). The mobile phase consisted of methanol (A), MTBE (B), and ultrapure water (C). The separation of analytes was performed with the following gradient: 0 min, 26% B; 0–10 min, 76% B; 10–14 min, 90% B; 14–16 min, 26% B; and 16–20 min, 26% B. The percent of C remained constant at 4% throughout the chromatographic run. The mobile phase flow was 0.4 mL min^−1^. The column temperature was 10 °C and the injection volume 5 µL. The autosampler temperature was maintained constant at 15 °C. Carotenoids were identified and quantified based on comparisons of their retention times and absorbance values of detected peaks from carrots with those obtained from the injection of each pure standard. To further verify peak identification and the absence of interferences at the analytes’ retention times, carrot samples were spiked with known concentrations of target compounds. Quantification was performed by means of an external calibration prepared with pure standards of each carotenoid (Sigma Aldrich, Atlanta, GA, USA) as described previously [43].

### 4.6. Antioxidant Capacity

Four different methods were used to estimate antioxidant capacity in the carrot extracts, namely the ‘2,2′-azino-bis(3-ethylbenzothiazoline-6-sulfonic acid’ (ABTS) [44], ‘2,2-diphenyl-1-picrylhydrazyl’ (DPPH) [45], ‘ferric reducing ability of plasma’ (FRAP) [46], and ‘oxygen radical absorbance capacity’ (ORAC) [47] assays. The ABTS and FRAP assays were used to estimate antioxidant capacity in carrot samples from both years, whereas DPPH and ORAC were used in samples from 2018. 

For the ABTS assay, the ABTS^⋅^^+^ working solution was prepared by mixing, in equal volumes, a 7.4 mM ABTS^⋅^^+^ solution and a 2.6 mM potassium persulfate solution. Both reagents were from Cayman Chemical Co. (Ann Arbor, MI, USA). The mixture was allowed to react for 12 h at room temperature in the dark. The solution was then diluted with methanol to obtain an absorbance of 0.700 ± 0.02 at 734 nm. A fresh dilute ABTS^⋅^^+^ solution was prepared for each assay. Carrot extracts (20 μL) were allowed to react with 980 mL of the ABTS^⋅+^ solution for 6 min in the dark. Then, absorbance readings were taken at 734 nm using a T60 Visible Spectrophotometer (PG Instruments, Leicestershire, United Kingdom). A calibration curve was prepared using Trolox (Cayman Chemical Co., Michigan, USA) as standard. Values were expressed as mmol of Trolox per kg fw.

The ability to reduce ferric ions (FRAP assay) was estimated using the original protocol [46] with modifications proposed by Locatelli et al. [48]. Briefly, an aliquot of 1 mL of the carrot extract was mixed with 1 mL of phosphate buffer (0.2 M) pH 6.6 and 1 mL of potassium ferricyanide (1% *w*/*v*). The mixture was incubated at 50 °C for 20 min. Finally, 1 mL of trichloroacetic acid was added. The mixture was centrifuged at 15,900× *g* for 10 min at 4 °C. The supernatant (1.5 mL) was mixed with 0.3 mL ferric chloride (1% *w*/*v*), and 1.5 mL of deionized water was added. After 10 min, absorbance readings at 700 nm were recorded in the same spectrophotometer apparatus described above. A calibration curve was prepared using Trolox as standard. FRAP values were expressed as mmol of Trolox per kg fw.

For the DPPH assay, the original protocol [45] was used with minor modifications. Briefly, a DPPH working solution was prepared by dissolving 40 mg of DPPH reactant per 1 L of methanol. Carrot extracts (50 μL) were allowed to react with 1000 µL of the DPPH solution in the dark for 1 h. Then, absorbance readings at 515 nm were recorded using a T60 Visible Spectrophotometer (PG Instruments, Leicestershire, United Kingdom). A calibration curve was prepared using Trolox as standard. The values were expressed as mmol of Trolox per kg fw. 

For the ORAC assay, the carrot extracts were diluted 1:750 *v*/*v* in 75 mM potassium phosphate buffer (pH 7.0). Aliquots (50 µL) of the diluted samples and Trolox standards were added to a 96-well black-colored plate. Then, 100 µL of fluorescein (20 nM solution) were added to the mixture and followed by incubation at 37 °C for 7 min before the addition of 50 µL of the peroxyl radical generator AAPH [2,2′-azobis(2-amidinopropane) dihydrochloride (Sigma-Aldrich Inc., Saint Louis, MO, USA), 140 mM solution]. Fluorescence was monitored using 485 nm excitation 154 and 538 nm emissions at 1 min intervals for 90 min on a microplate fluorometer (Fluoroskan 155 Ascent FL, Thermo Fisher Scientific Inc., Wilmington, DE, USA). The area under the curve of the fluorescence decay during 90 min was calculated and the ORAC was expressed as mmol of Trolox per kg fw. 

### 4.7. Data Analysis

The data were evaluated using analysis of variance (ANOVA), and the means were compared by the least significant difference test (LSD Fisher), considering significant *p* values ≤ 0.05. Correlation analyses were performed using the Spearman correlation coefficient. The InfoStat statistical software [49] was used for all the analyses.

### 4.8. Principal Component Analysis

Principal component analysis (PCA) was implemented in the InfoStat software to classify the cultivars based on their bioactive compounds and antioxidant activities. The data set consisted of a matrix of the order 29 × 16, where the rows represent the 29 carrot accessions and the columns comprised the data for the root concentration of 12 phytochemicals [five anthocyanin pigments (Cy3XG, Cy3XGG, Cy3XSGG, Cy3XFGG, and Cy3XCGG), total acylated anthocyanins (TAA), total non-acylated anthocyanins (TNAA), total anthocyanin content by HPLC-UV-Vis (TAC_hplc_) and spectrophotometry (TAC_spec_), total phenolics (TPC), and two carotenoids (β-carotene and lutein)] and antioxidant capacity estimated by four analytical methods (ABTS, FRAP, DPPH, and ORAC). The data for three biological replicates were expressed as means.

## 5. Conclusions

Carrot anthocyanins are of high interest for increasing nutraceutical value in this crop, due to the alleged health-enhancing properties of these pigments, and for their use as natural food colorants. The characterization, over two years, of this purple carrot collection (and two non-purple carrots) for concentration of anthocyanins, phenolics, and carotenoids, and antioxidant capacity by four different methods, revealed fairly consistent data across years and methods, indicating a significant and broad variation among the accessions for all the traits analyzed. In general, carrots with intense purple pigmentation in all the root tissues exhibited the greatest anthocyanin and phenolic contents, and the strongest antioxidant capacities, measured by all methods. This observation, and the moderate –to strong positive correlations found between the content of all individual anthocyanins’ pigments, TAC, and TPC with antioxidant capacity (r = 0.59–0.90), strongly suggest that anthocyanins and other phenolics are the most important contributors to antioxidant capacity in purple carrots. In contrast, carotenoid pigments were not positively associated with antioxidant capacity. 

In all of the accessions, AA predominated over NAA (AA represented ≥55.5% of total anthocyanins), with pigments acylated with ferulic and coumaric acids being the most abundant carrot anthocyanins. We identified accessions with high concentrations of chemically stable AA, some of which were particularly rich in Cy3XFGG, considered to be the most stable carrot anthocyanin. These materials are most suitable for the production of chemically stable food dyes. Despite the fact that AA prevailed over NAA in all of the accessions, a few materials had a relatively high NAA content (up to 800 mg kg^−1^ fw), which are of value as sources of highly bioavailable pigments, for example, for fresh consumption of carrots. Altogether, it is expected that these data will be useful to breeding programs aiming at increasing carrots’ nutritional value, as well as to consumers trying to improve their health through informed dietary choices.

## Figures and Tables

**Figure 1 plants-12-01796-f001:**
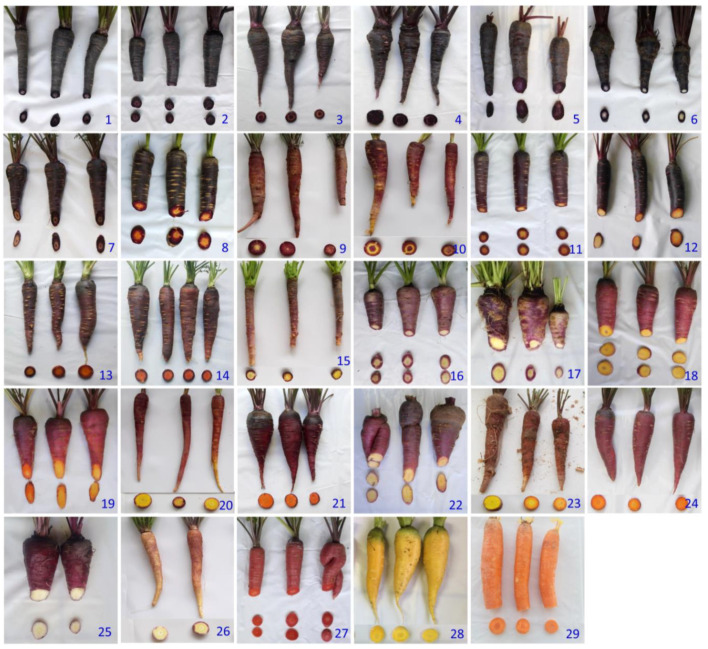
Root phenotypes for 29 carrot accessions used in the study. Accession numbers refer to the carrot materials described in Table 1.

**Figure 2 plants-12-01796-f002:**
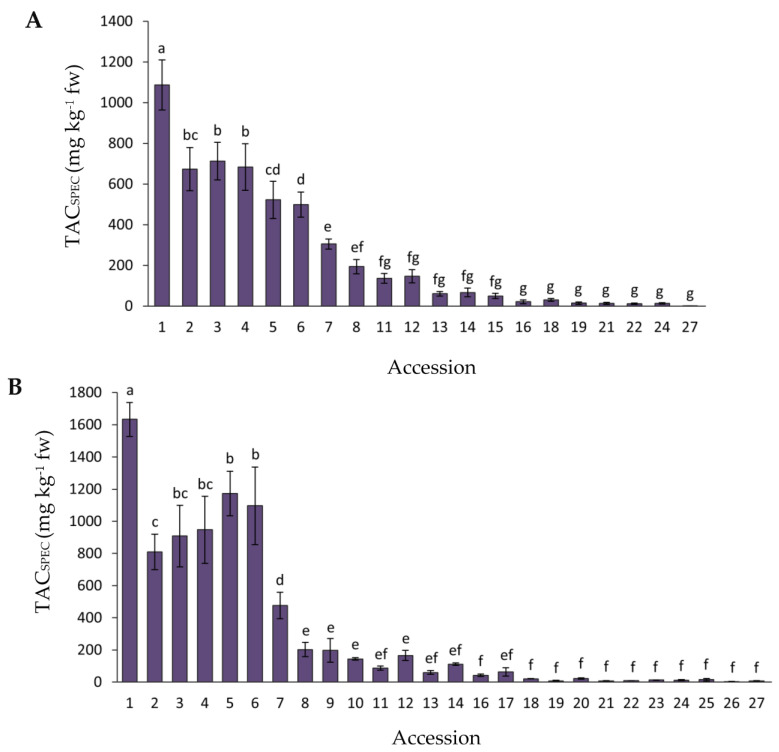
Variation for root total anthocyanin content, estimated by spectrophotometry (TAC_SPEC_), among 27 purple-rooted carrot accessions, in 2018 (**A**) and 2019 (**B**). Accession numbers refer to the carrot materials described in Table 1 and are presented in decreasing order according to their total anthocyanin content, as evaluated by HPLC analysis in 2018. Bars indicate mean values of three replicates, expressed as mg of cyanidin equivalents per kg of fresh weight (mg kg^−1^ fw) ± standard error. Mean values with a common letter are not significantly different at *p* ≤ 0.05 (LSD Fisher test).

**Figure 3 plants-12-01796-f003:**
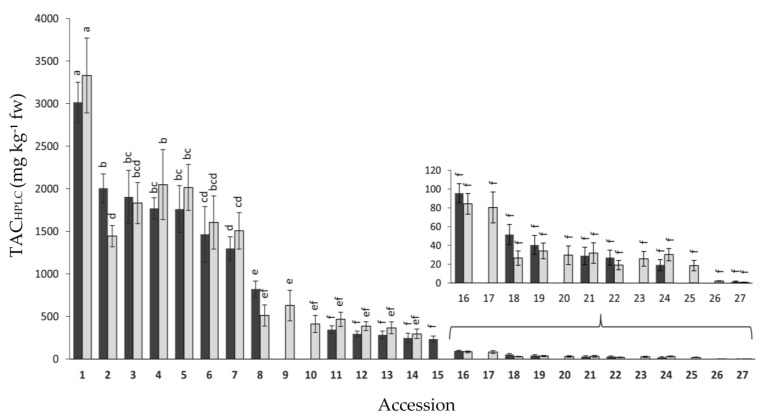
Variation for root total anthocyanin content, estimated by HPLC analysis (TAC_HPLC_), among 27 purple-rooted carrot accessions, in 2018 (black bars) and 2019 (gray bars). Accession numbers refer to the carrot materials described in Table 1 and are presented in decreasing order according to their total anthocyanin content, as evaluated by HPLC analysis in 2018. Bars indicate mean values of three replicates, expressed as mg of cyanidin equivalents per kg of fresh weight (mg kg^−1^ fw) ± standard error. Mean values with a common letter are not significantly different at *p* ≤ 0.05 (LSD Fisher test), regardless of year. No significant differences were found between years. The sub-figure in the bottom right depicts variation among the accessions with less than 150 mg/kg fw of total anthocyanins.

**Figure 4 plants-12-01796-f004:**
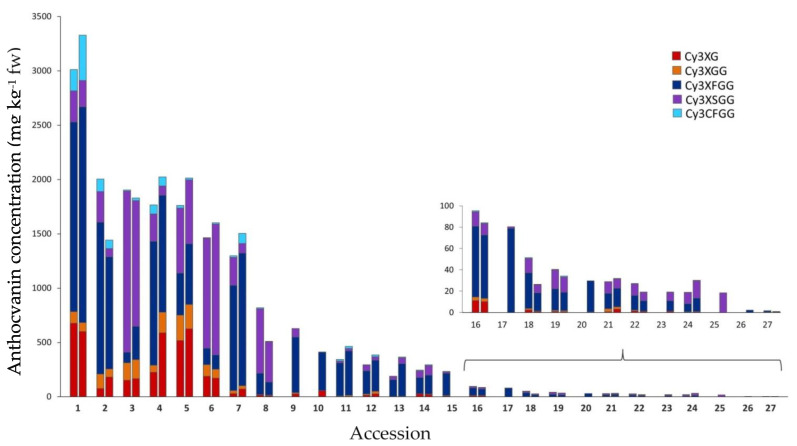
Root anthocyanin composition in 27 purple carrot accessions. For each accession, the left and right bars indicate carrots grown in 2018 and 2019, respectively. Accession numbers refer to the carrot materials described in Table 1 and are presented in decreasing order according to their total anthocyanin content, as evaluated by HPLC analysis in 2018. Bars for total anthocyanin concentration content were partitioned into their mean content of individual anthocyanin pigments, expressed as mg kg^−1^ fw. The sub-figure in the bottom right depicts anthocyanin profiles in accessions with less than 150 mg/kg fw of total anthocyanins. Acylated and non-acylated anthocyanins are indicated in cold (light blue, violet, and dark blue) and warm colors (red and orange), respectively. Cy3XG. Cyanidin-3-(2″-xylose-galactoside); Cy3XGG. Cyanidin-3-(2″-xylose-6-glucose-galactoside); Cy3XFGG. Cyanidin-3-(2″-xylose-6″-feruloyl-glucose-galactoside); Cy3XSGG. Cyanidin-3-(2″-xylose-6″-sinapoyl-glucose-galactoside); Cy3XCGG. Cyanidin-3-(2″-xylose-6″-(4-coumaroyl)glucose-galactoside).

**Figure 5 plants-12-01796-f005:**
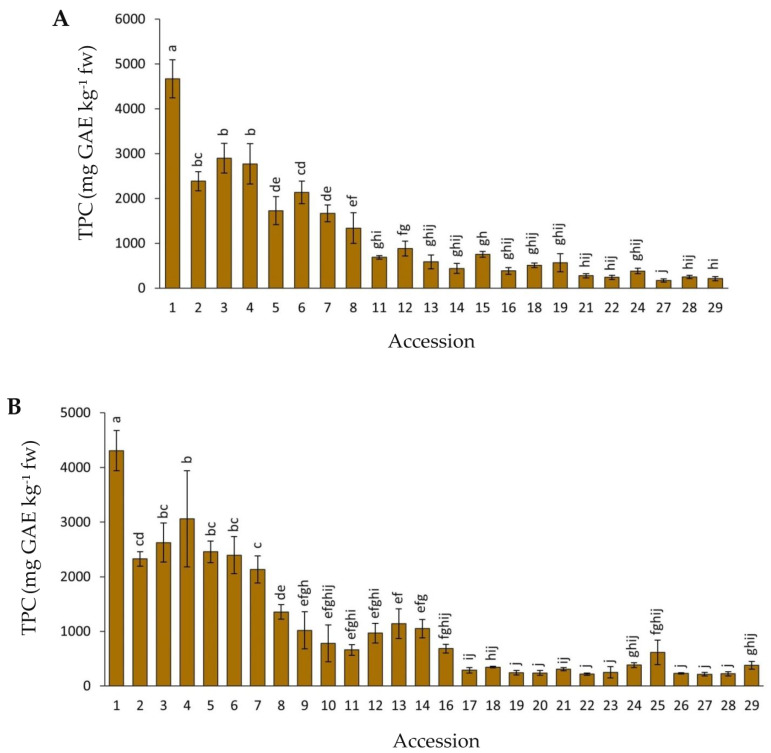
Root total phenolics content (TPC) for carrot accessions grown in 2018 (**A**) and 2019 (**B**). Accession numbers refer to the carrot materials described in Table 1. Bars indicate mean values of three replicates, expressed as mg of gallic acid equivalents (GAE) per kg of fresh weight (mg GAE kg^−1^ fw) ± standard error. Mean values with a common letter are not significantly different at *p* ≤ 0.05 (LSD Fisher test).

**Figure 6 plants-12-01796-f006:**
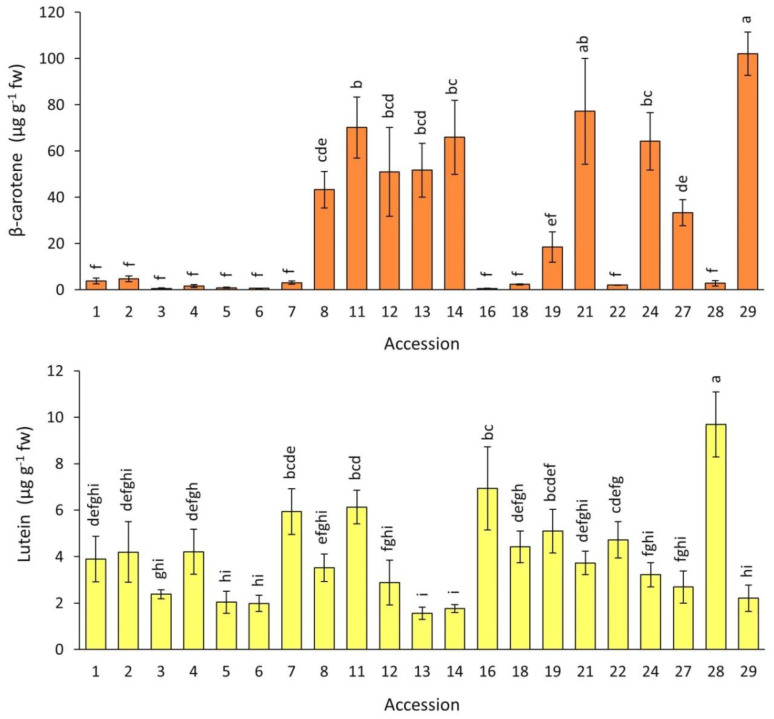
Root β-carotene and lutein content in carrot accessions field-grown in 2018. Accession numbers refer to the carrot materials described in Table 1. Bars indicate mean values of three replicates, expressed as µg g^−1^ fw ± SE. Mean values with a common letter are not significantly different at *p* ≤ 0.05 (LSD Fisher test).

**Figure 7 plants-12-01796-f007:**
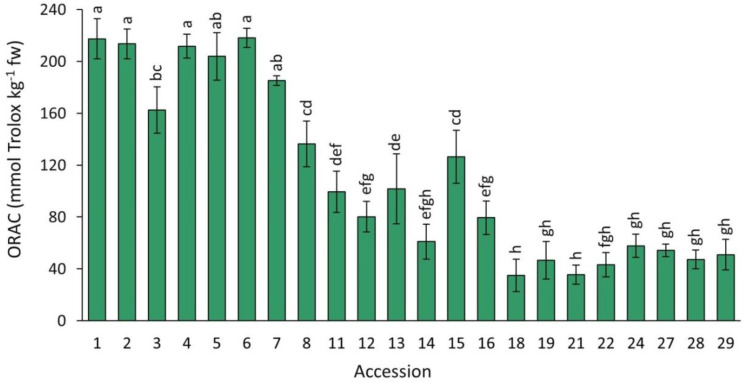
Antioxidant capacity by ORAC in carrot accessions grown in 2018. Accession numbers refer to the carrot materials described in Table 1. Bars indicate mean values of three replicates, expressed as mmol Trolox kg^−1^ fw ± SE. Mean values with a common letter are not significantly different at *p* ≤ 0.05 (LSD Fisher test).

**Figure 8 plants-12-01796-f008:**
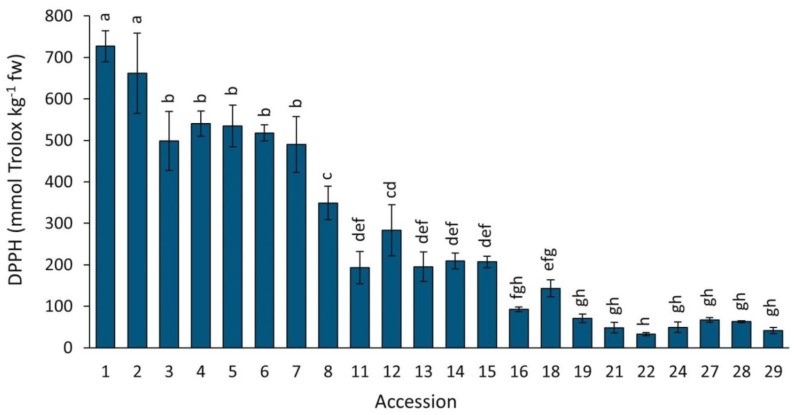
Antioxidant capacity by DPPH in carrot accessions grown in 2018. Accession numbers refer to the carrot materials described in Table 1. Bars indicate mean values of three replicates, expressed as mmol Trolox kg^−1^ fw ± SE. Mean values with a common letter are not significantly different at *p* ≤ 0.05 (LSD Fisher test).

**Figure 9 plants-12-01796-f009:**
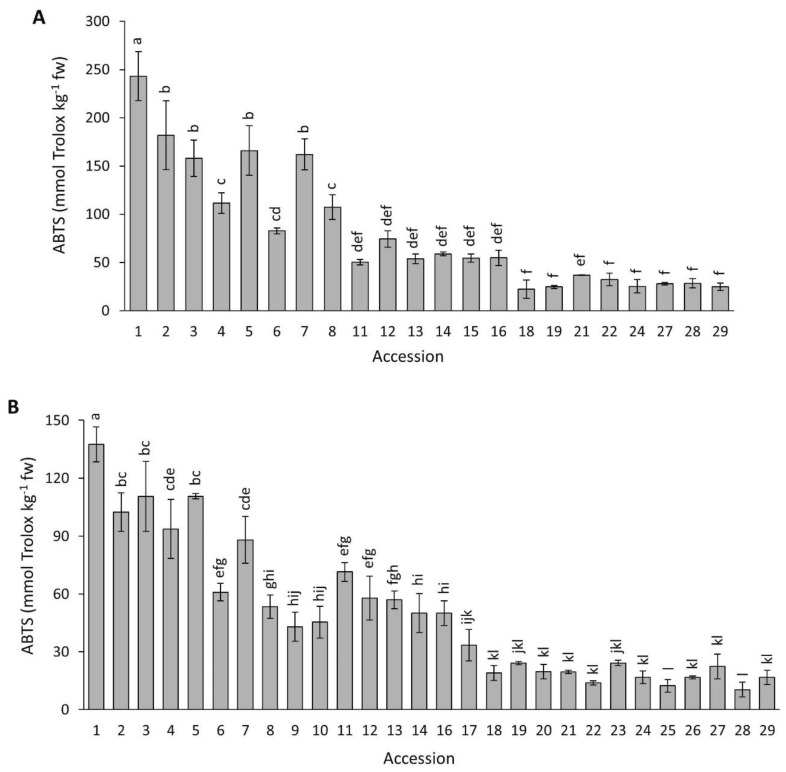
Antioxidant capacity by ABTS in carrot accessions grown in 2018 (**A**) and 2019 (**B**). Accession numbers refer to the carrot materials described in Table 1. Bars indicate mean values of three replicates, expressed as mmol Trolox kg^−1^ fw ± SE. Mean values with a common letter are not significantly different at *p* ≤ 0.05 (LSD Fisher test).

**Figure 10 plants-12-01796-f010:**
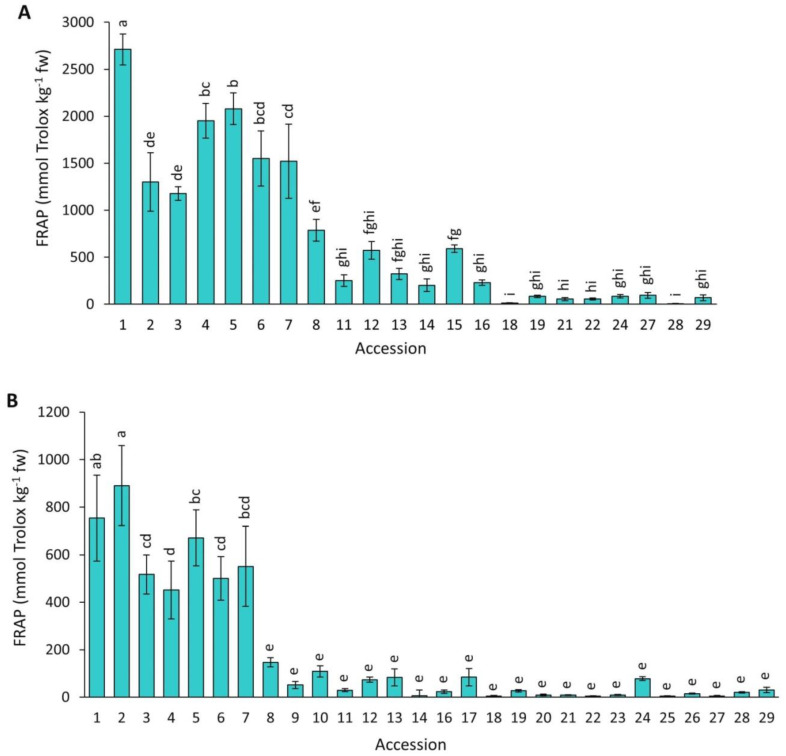
Antioxidant capacity by FRAP in carrot accessions grown in 2018 (**A**) and 2019 (**B**). Accession numbers refer to the carrot materials described in Table 1. Bars indicate mean values of three replicates, expressed as mmol Trolox kg^−1^ fw ± SE. Mean values with a common letter are not significantly different at *p* ≤ 0.05 (LSD Fisher test).

**Figure 11 plants-12-01796-f011:**
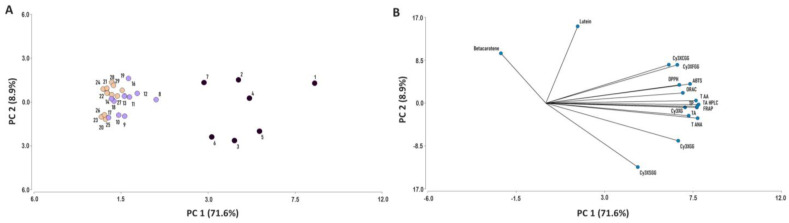
Score plot (**A**) and loading plot (**B**) of the principal component analysis (PCA) for 12 phytochemicals [five anthocyanin pigments (Cy3XG, Cy3XGG, Cy3XSGG, Cy3XFGG, and Cy3XCGG), total acylated anthocyanins (TAA), total non-acylated anthocyanins (TNAA), total anthocyanin content by HPLC (TA HPLC), total anthocyanins by spectrophotometry (TA), total phenolics (TPC), β-carotene, and lutein] and antioxidant capacity estimated by four analytical methods (ABTS, FRAP, DPPH, and ORAC), for 29 carrot accessions. Data from years 2018 and 2019 were combined. The numbers of the accessions in the score plot refer to the carrot materials described in Table 1, indicating with black, violet, and orange circles accessions with high (>1000 mg kg^−1^ fw), intermediate (80–1000 mg kg^−1^ fw), and low anthocyanin content (<80 mg kg^−1^ fw), respectively. In the loading plot, the lines starting from the center point of the bi-plot depict the positive or negative association of the parameters with the two principal components.

**Table 1 plants-12-01796-t001:** Main characteristics of 29 carrot accessions used in this study.

Reference Number ^ζ^	Plant/Cultivar Name	Accession ID	Genetic Structure	Petiole and Root Olor Phenotype ^§^	Seed Source and Location	Geographic Origin *
1	P9547	PI 167055	inbred line	P-PPPP	GRIN-USDA, Beltsville, MD, USA	Eregli/Hatay, Turkey
2	Purple 68		F1	P-PPPP	Territorial Seed Company, Cottage Grove, OR, USA	NA
3	Pusa asita		OP	P-PPPP	Baker Creek Heirloom Seeds, Mansfield, MO, USA	India
4	Night bird		F1	P-PPPP	Plant World Seeds, Newton Abbot, UK	NA
5	INTA43		OP	P-PPPP	INTA La Consulta, Mendoza, Argentina	NA
6	Black nebula		OP	P-PPPW/P-PPWW	Sustainable Seed Company, Chico, CA, USA	NA
7	Black carrot		F1	P-PPPP/P-PPYP	Sedi Seeds Co., Singapore	NA
8	B7262		inbred line	G-PPOO	USDA-ARS, Madison, WI, USA	Turkey
9	Dragon		OP	P-PPPP/P-PPPY	Territorial Seed Company, Cottage Grove, OR, USA	NA
10	340	PI 167143	OP	P-PPYP	GRIN-USDA, Beltsville, MD, USA	Mersin, Turkey
11	Purplesnax		F1	G-PPOO	Territorial Seed Company, Cottage Grove, OR, USA	NA
12	Purple carrot		OP	P-PPOO/P-PPYY	Xiangqutao Store, Singapore	NA
13	Purple elite		F1	G-PPOO	Stokes Seeds, Thorold, ON, Canada	NA
14	Purple haze		F1	G-PPOO	Territorial Seed Company, Cottage Grove, OR, USA	NA
15	Spanish black		OP	G-PPYY	Magic Garden Seeds, Regensburg, Germany	NA
16	Gniff		OP	G-PPYY/G-PPWW	Baker Creek Heirloom Seeds, Missouri, USA	Tessin, Switzerland
17	1540	PI 223361	landrace	G-PPYY/G-PPWW	GRIN-USDA, USA	Ardabil, Azerbaijan
18	Ping Ding	PI 652188	OP	P-PYYY	GRIN-USDA, USA	China
19	Homs	BP85682	OP	P-POOO	USDA-ARS, Wisconsin, USA	Homs, Syria
20	Havuc	PI 167211	OP	P-PPYY/P-PYYY	GRIN-USDA, USA	Mersin, Turkey
21	Purple dragon		OP	P-POOO	Baker Creek Heirloom Seeds, Missouri, USA	NA
22	INTA45		OP	P-PYYY	INTA La Consulta, Mendoza, Argentina	China
23	IIHR 189	PI 652252	landrace	P-PYYY/P-POOO/P-POOY	GRIN-USDA, USA	Uttar Pradesh, India
24	INTA44		inbred line	P-POOO	INTA La Consulta, Mendoza, Argentina	Syria
25	Zardak tabur	PI 254552	landrace	P-PWWW	GRIN-USDA, USA	Kābul, Afghanistan
26	Nargesi Shiraz	PI 226636	landrace	P-PWW	GRIN-USDA, USA	Fārs, Iran
27	Malbec		F1	P-PRRR/P-PPRR	Stokes Seeds, Canada	NA
28	Yellowstone		OP	G-YYYY	Suttons Seeds, Paignton, UK	NA
29	Autumn king		OP	G-OOOO	Seed Parade Co., Isleworth, UK	NA

**^ζ^** Reference numbers of the carrot accessions are based on their ranks for total anthocyanin concentration as estimated by HPLC analysis in 2018. F1 = hybrid, OP. open-pollinated. ^§^ The order of phenotype coding for different plant tissues is as follows: leaf petiole—root periderm—outer phloem—inner phloem—xylem; where petiole color was either purple (P) or green (G), and the root tissues were purple (P), orange (O), yellow (Y), white (W), or red (R). * Geographic origin of the original population from which the cultivar/line was developed. NA = Data not available.

**Table 2 plants-12-01796-t002:** Carrot cyanidin derivatives with approximate HPLC retention times and molecular masses.

Compound	Abbreviation	RT ^ζ^	MW ^§^
Cyanidin-3-(2″-xylose-galactoside)	Cy3XG	14.0	581
Cyanidin-3-(2″-xylose-6-glucose-galactoside)	Cy3XGG	13.4	743
Cyanidin-3-(2″-xylose-6″-sinapoyl-glucose-galactoside)	Cy3XSGG	14.3	949
Cyanidin-3-(2″-xylose-6″-feruloyl-glucose-galactoside)	Cy3XFGG	14.8	919
Cyanidin-3-(2″-xylose-6″-(4-coumaroyl)glucose-galactoside)	Cy3XCGG	15.2	889

**^ζ^** RT is the approximate retention time (min) for the chromatographic procedure described in the Section 4. ^§^ MW is molecular weight.

**Table 3 plants-12-01796-t003:** Pairwise Pearson correlation coefficient values (r) among carrot bioactive compounds and antioxidant capacity for years 2018 and 2019.

	TAC_SPEC_	TAC_HPLC_	Cy3XG	Cy3XGG	Cy3XSGG	Cy3XFGG	Cy3XCGG	TAA	TNAA	TPC	ABTS	FRAP	DPPH	ORAC	β-Carotene	Lutein
TAC_SPEC_		0.96 ***	0.88 ***	0.92 ***	0.89 ***	0.87 ***	0.86 ***	0.96 ***	0.93 ***	0.94 ***	0.89 ***	0.86 ***	0.72 ***	0.85 ***	−0.36 *	−0.04
TAC_HPLC_	0.94 ***		0.89 ***	0.94 ***	0.92 ***	0.91 ***	0.90 **	0.99 ***	0.94 ***	0.92 ***	0.90 ***	0.88 ***	0.72 ***	0.87 ***	−0.38 *	−0.08
Cy3XG	0.86 ***	0.89 ***		0.88 ***	0.82 ***	0.79 ***	0.76 ***	0.86 ***	0.97 ***	0.80 ***	0.82 ***	0.84 ***	0.66 ***	0.78 ***	−0.48 ***	−0.16
Cy3XGG	0.77 ***	0.82 ***	0.84 ***		0.85 ***	0.84 ***	0.81 ***	0.93 ***	0.95 ***	0.88 ***	0.87 ***	0.84 ***	0.65 ***	0.83 ***	−0.45 ***	−0.07
Cy3XSGG	0.77 ***	0.82 ***	0.71 ***	0.82 ***		0.75 ***	0.74 ***	0.91 ***	0.86 ***	0.85 ***	0.82 ***	0.80 ***	0.66 ***	0.77 ***	−0.33 **	−0.18
Cy3XFGG	0.89 ***	0.94 ***	0.85 ***	0.74 ***	0.67 ***		0.95 ***	0.92 ***	0.84 ***	0.82 ***	0.83 ****	0.81 ***	0.64 ***	0.78 ***	−0.23	−0.01
Cy3XCGG	0.76 ***	0.83 ***	0.82 ***	0.88 ***	0.74 ***	0.79 ***		0.91 ***	0.81 ***	0.81 ***	0.80 ***	0.75 ***	0.66 ***	0.73 ***	−0.21	0.03
TAA	0.94 ***	0.99 ***	0.87 ***	0.81 ***	0.82 ***	0.94 ***	0.82 ***		0.92 ***	0.92 ***	0.90 ***	0.87 ***	0.72 ***	0.87 ***	−0.36 **	−0.05
TNAA	0.87 ***	0.91 ***	0.99 ***	0.89 ***	0.75 ***	0.86 ***	0.85 ***	0.89 ***		0.86 ***	0.88 ***	0.87 ***	0.69 ***	0.82 ***	−0.48 ***	−0.12
TPC	0.86 ***	0.88 ***	0.82 ***	0.79 ***	0.82 ***	0.79 ***	0.78 ***	0.88 ***	0.84 ***		0.86 ***	0.83 ***	0.66 ***	0.86 ***	−0.39 *	−0.01
ABTS	0.68 ***	0.71 ***	0.72 ***	0.85 ***	0.73 ***	0.63 ***	0.87 ***	0.70 ***	0.76 ***	0.77 ***		0.87 ***	0.75 ***	0.86 ***	−0.36 **	−0.07
FRAP	0.76 ***	0.79 ***	0.68 ***	0.65 ***	0.59 ***	0.73 ***	0.65 ***	0.79 ***	0.70 ***	0.78 ***	0.64 ***		0.66 ***	0.85 ***	−0.43 ***	−0.17
DPPH	-	-	-	-	-	-	-	-	-	-	-	-		0.66 ***	−0.23	0.01
ORAC	-	-	-	-	-	-	-	-	-	-	-	-	-		−0.47 ***	−0.06
β-carotene	-	-	-	-	-	-	-	-	-	-	-	-	-	-		0.01
Lutein	-	-	-	-	-	-	-	-	-	-	-	-	-	-	-	

The diagonal gray boxes separate data for 2018 (upper half) and 2019 (lower half). TAC_SPEC_: total anthocyanin content by spectrophotometry; TAC_HPLC_: total anthocyanin content by HPLC; TAA: total acylated anthocyanins; TNAA: total non-acylated anthocyanins; TPC: total phenolics content. The full names of individual anthocyanin pigments (Cy3XG, Cy3XGG, Cy3XSGG, Cy3XFGG, Cy3XCGG) are presented in Table 2. Antioxidant capacity was estimated by four analytical methods (ABTS, FRAP, DPPH, and ORAC). *, **, *** indicate significant correlation at *p* < 0.05, *p* < 0.01, and *p* < 0.001, respectively.

## Data Availability

All data are included in the main text and Appendix A.

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
