# Peer review of "Characterization of Purple Carrot Germplasm for Antioxidant Capacity and Root Concentration of Anthocyanins, Phenolics, and Carotenoids"

_plants, 2023, doi:10.3390/plants12091796_

Round 1

Reviewer 1 Report

The manuscript by María B. Pérez et al. entitled “Characterization of purple carrot germplasm for antioxidant capacity and root concentration of anthocyanins, phenolics, and carotenoids”

Even if not too original, the manuscript could be of some interest for the readers of the Journal. The technical aspects of the presented data seem to be good, and the analytical work were properly performed. In the Discussion section the authors provide a good presentation of their results. The manuscript is written in a good English. For these reasons, in my opinion the manuscript is suitable for the publication in Plants.

Reviewer 2 Report

Overall, this is a well-designed and informative study that characterizes a diverse collection of purple and non-purple carrots for their antioxidant capacity and concentration of anthocyanins, phenolics, and carotenoids. The use of multiple methods to estimate antioxidant capacity is a strength of the study, as is the consideration of both genetically and phenotypically diverse carrot accessions. However, there are some areas where the manuscript could be improved: The abstract provides a good summary of the study, but could be clearer on the specific findings and implications. For example, it would be helpful to explicitly state that the study found a positive correlation between antioxidant capacity and anthocyanin and phenolic content, but not carotenoid content. The methods section could benefit from more detailed information on the growing conditions and sampling procedures. For example, were the carrots grown under the same conditions for both years? How were the samples prepared for analysis? While the discussion section provides some valuable insights into the potential applications of the findings (e.g., identification of accessions with high concentrations of stable anthocyanins for food dye production), it could be expanded to provide more context and comparison to previous studies in the field. The use of multiple statistical tests and correlation analyses is appropriate, but it would be helpful to provide effect size estimates (e.g., Cohen's d) to quantify the magnitude of differences between groups and the strength of relationships between variables. Overall, this study provides valuable insights into the diversity and potential nutritional and industrial applications of purple carrot germplasm, but could benefit from some minor revisions to improve clarity and provide additional context.

Reviewer 3 Report

The article by Belén Pérez et al. concerns with the definition of  antioxidant capacity and chemical profile of several genotypes of purple carrots. 

While the study was well conducted, there are several improvements to be performed regarding the statistical analysis:

1)  The authors should better state the true number of individual samples: 29 or 29x3?

2) The PCA in figure 11 should be divided in a panel with on the left the score plot and on the right the loading plot, moreover the scores should be colored according to  the genotype;

3) Figure 3 is very interesting, but, due to the nature of Pearson's correlations, they are better employed to highlight the covariance of the variables belonging to a single class, i.e. genotype. Did the authors try to recalculate the matrix with a limited selection of genotypes? Moreover, was the structure of the covariances affected by the year of sampling? Please, add the critical rho value in the caption 

Round 2

Reviewer 3 Report

The authors improved the manuscript and fulfilled all the reviewers' requests, so the article can be accepted in the present form.